# The silent majority: The typical Canadian sex worker may not be who we think

**Lynn Kennedy** [ORCID] *

Sex Work Population Project

* lynn.kennedy@populationproject.ca

## Abstract

### Background

Most sex worker population studies measure population at discrete points in time and very few studies have been done in industrialized democracies. The purpose of this study is to consider how time affects the population dynamics of contact sex workers in Canada using publicly available internet advertising data collected over multiple years.

### Methods

3.6 million web pages were collected from advertising sites used by contact sex workers between November, 2014 and December, 2016 inclusive. Contacts were extracted from ads and used to identify advertisers. First names were used to estimate the number of workers represented by an advertiser. Counts of advertisers and names were adjusted for missing data and overcounting. Two approaches for correcting overcounts are compared. Population estimates were generated weekly, monthly and for the two year period. The length of time advertisers were active was also estimated. Estimates are also compared with related research.

### Results

Canadian sex workers typically advertised individually or in small collectives (median name count 1, IQR 1–2, average 1.8, SD 4.4). Advertisers were active for a mean of 73.3 days (SD 151.8, median 14, IQR 1–58). Advertisers were at least 83.5% female. Respectively the scaled weekly, monthly, and biannual estimates for female sex workers represented 0.2%, 0.3% and 2% of the 2016 Canadian female 20–49 population. White advertisers were the most predominant ethnic group (53%).

### Conclusions

Sex work in Canada is a more pervasive phenomenon than indicated by spot estimates and the length of the data collection period is an important variable. Non-random samples used in qualitative research in Canada likely do not reflect the larger sex worker population represented in advertising. The overall brevity of advertising activity suggests that workers typically exercise agency, reflecting the findings of other Canadian research.

**Editor:** Hamid Sharifi, HIV/STI Surveillance Research Center and WHO Collaborating Center for HIV Surveillance, Institute for Future Studies in Health, Kerman University of Medical Sciences, ISLAMIC REPUBLIC OF IRAN

**Data Availability Statement:** Data may be found at https://osf.io/mebvp/.

**Funding:** The authors received no specific funding for this work.

**Competing interests:** The authors have declared that no competing interests exist.

## Introduction

The purpose of this study is to provide a more comprehensive picture of the population dynamics of contact sex workers in Canada by examining sex worker advertising behavior over multiple years. How does the population change over time? Most importantly, can we say that all Canadian sex workers are represented in the debate around policy and in the research that is often used as the basis for policy?

The majority of studies that estimate sex worker populations on a national scale only attempt to generate estimates at a single point in time. These studies nevertheless are usually costly, large-scale efforts that use a variety of techniques. These techniques include: in-person interviews [1–9]; respondent driven and token based sampling (where researchers use an initial group of participants to recruit other participants) [4–6]; in-person site counts (where researchers identify geographic "hotspots" where a population of interest is known to frequent, typically by recruiting and interviewing local experts, and visually enumerate the people at that location over a preselected time period) [3,4,6–8]; and indirect mapping via service delivery statistics (where non-governmental organization (NGO) or police statistics are used as a basis to infer population) [3,7,9–11]. One study from New Zealand combined counts of street workers with in person enumeration based on newspaper and internet advertising [12]. With the exception of Abel et al. [12], female sex workers (FSW) or female identified sex workers are the subject of the research. Most of the research is limited to populations in the Global South.

A number of criticisms of existing population studies have been put forward. Both Cusick et al. [11] and Abel et al. [12] describe how population estimates can be difficult to confirm and can inflate numbers when based on NGO and police records as these are often kept long after workers have left the industry. Population estimates that depend on specific locales or social networks [3–6,13] can leave out workers who are not part of these contexts. Workers who travel, for example, or who are only in the industry for short periods of time may be excluded.

Population studies are not the only research affected by sampling methodology. Many Canadian qualitative studies of sex workers use non-random samples [14–24]. It is an open question whether non-random samples typically used in this research accurately reflect the demographic composition of the populations they intend to represent. This is an important question as this research is often used as the basis for government policy.

Internet advertising has become the dominant form of advertising for sex workers in Canada. Internet-based advertising has been shown to be a powerful tool that can significantly improve risk-management, safety, and communication between sex workers and clients. Research has shown that online communication has allowed sex workers to communicate significant details to clients including their preferences and personal health practices [14,25].

This study shows that population estimates can be made at a much lower cost compared to other methods as data collection and analysis are mostly automated. The source records, created by the advertisers themselves, can be extensive, improving the completeness of estimates and including otherwise hidden populations. Perhaps most important, the length of time individuals are involved as contact sex workers can be estimated. While the majority appear to be in the industry for relatively brief periods of time, there is no "one size fits all" description that applies to all sex workers.

## Methods

### Overview

This study provides evidence for the number of sex workers advertising over specific time periods using publicly available internet advertising data. Data was collected and analyzed over a two year period using a combination of open source and custom software tools [26]. Fig 1

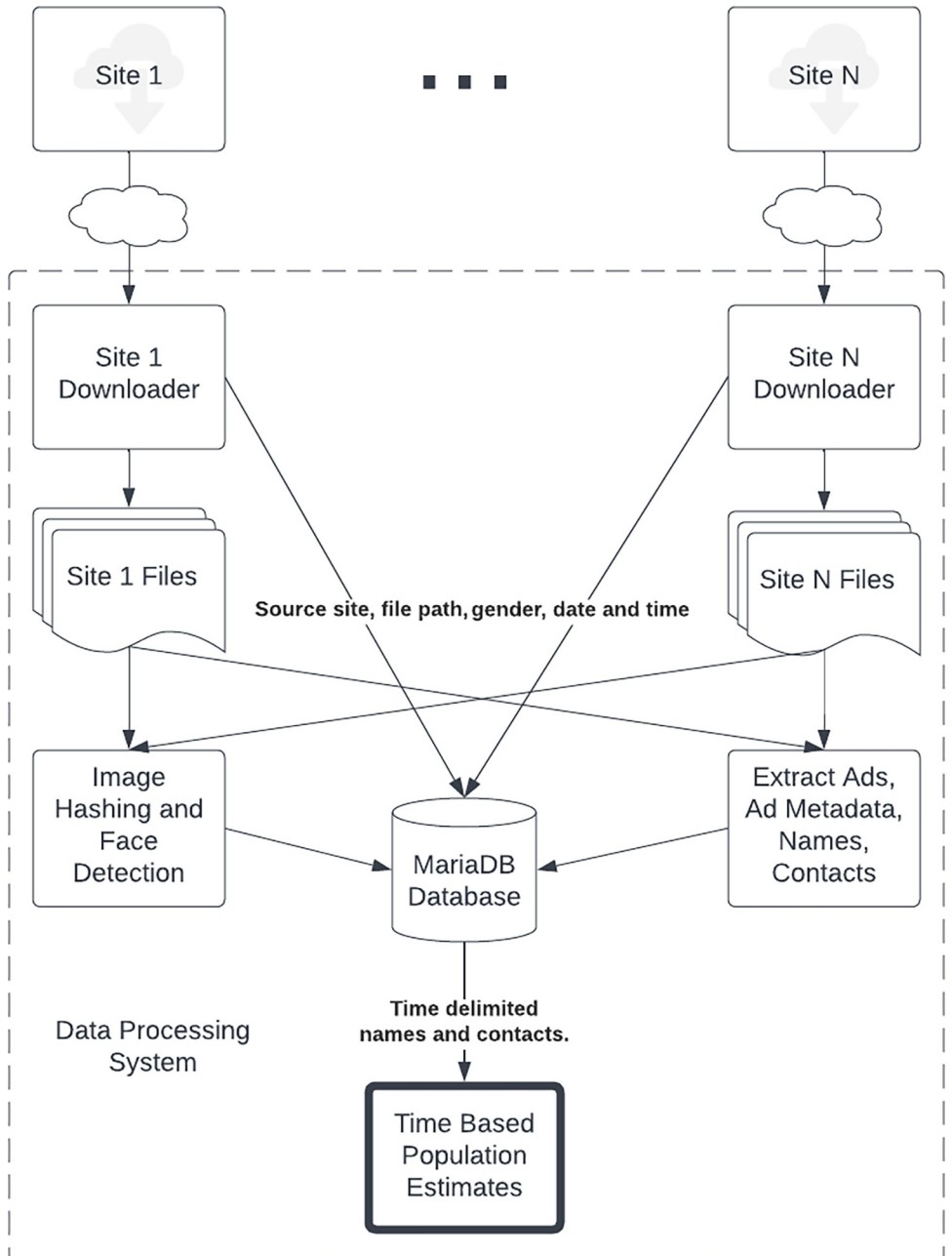

**Fig 1. Flow diagram illustrating the data processing pipeline used to generate population estimates.**

provides an overview of the processing pipeline and Table 1 outlines the steps taken to generate population statistics. This process was done in four main phases. The first phase involved downloading data from the target websites. Once files were downloaded, they were analyzed and grouped based on contact metadata. Names were detected and counted in ads to estimate the actual number of workers represented. Collected metadata was then analyzed for errors and finally statistics were generated. Most steps were done once but some, such as cluster analysis, name interpolation and scaling, were done on an ad hoc basis when analyzing population for a specific time period.

**Table 1. Data processing steps for population estimation.** Steps were automated except where indicated.

| Step | Description |
|---|---|
| Collect data | Custom download programs make local copies of web page files and associated images for further processing and analysis. Date, time and gender metadata are collected here. |
| Metadata extraction | |
| Analyze images | Images are hashed to identify related images and analyzed for faces. |
| Extract contacts | Find phone numbers and email addresses in ad text. |
| Extract names | Find first names in ad text. |
| Extract other metadata | Identify other variables of interest. Gender and social context (collective vs individual) are the main ones used in this analysis. |
| Cluster analysis | Analyze contacts to find those that co-occur in ads. Treat these related contacts as a single virtual contact. This is done as needed for a given time period. |
| Error estimation and mitigation | |
| Check for bad contact data* | The automated extraction process for contact data can pick up invalid contact information. Invalid contacts are either removed or combined with related contacts. |
| Estimate proportion of valid advertisers* | Not every advertiser is a contact sex worker. The number of relevant advertisers is counted from a sample of advertisers based on the criteria described in supplemental materials S1 File. This is used in the scaling calculation described below. |
| Estimate proportion of valid names* | For any given contact, names may have been extracted in error. For a random sample of advertiser-name pairs the number of correct pairs are counted. This is used in the scaling formula described below. |
| Image validation* | Check samples of images for the validity of image hashes. Determine the optimum confidence level for face detection. See S1 Appendix. |
| Estimate contact change rate | Some advertisers change contacts periodically. This can be measured from ad metadata and related images. The rate of contact change is used in the scaling calculation described below. |
| Estimate probability name is new | Some workers change their names. The probability that any name seen is in fact referring to a worker that has not been seen before is measured using methods similar to the contact change measure. |
| Interpolate missing name counts | When advertisers lack name data, add in name counts based on median values for individual and collective advertisers. This is done as needed for a given time period. |
| Apply scaling | A scaling calculation reduces the advertiser and worker counts based on the contact change rate and the proportions of valid names and contacts. This is done as needed for a given time period. |
| Measures and statistical analyses | |
| Generate population estimates | Sum raw, interpolated and scaled name and advertiser counts for multiple time periods. Generate descriptive statistics for shorter periods. Name counts are stratified by gender. |
| Identify trends | Create linear regression models for ad count, advertisers and workers versus month for the study period. |
| Days online | Generate descriptive statistics for the number of days online for advertisers. |
| Social context | Measure social context by comparing proportions of individual versus collective advertisers and workers. |

*indicates the step was not automated.

## Data collection

The sites analyzed in this study represent where the majority of sex work advertising occurred in Canada during the 2014–2016 study period according to advisors from the *Sex, Power, Agency, Consent, Environment and Safety Project* (SPACES) [24]. SPACES was initiated in 2012 at the University of British Columbia to explore health and safety issues experienced by

off-street sex workers. The SPACES advisors were people with experience in contact sex work either as workers or third parties who were users of such websites.

Each source website had a unique structure. Customized downloaders for each site were developed to gather ads and associated images ["downloaders" in 26]. Sites were checked for new ad pages at least every 15 minutes. It was assumed to be very unlikely that an advertiser would post and then delete an ad in less than 15 minutes.

For the purposes of time delimited population estimates, source data was restricted to classified sites that provided time-stamped ads from distinct advertisers where each ad was a unique web page. Content from other sources was analyzed separately for purposes of comparison.

## Metadata extraction

Ad text was isolated from each web page and was parsed for metadata [26]. The metadata used for the population estimates consisted of contact information, first names, gender and whether an advertiser represented a collective or an individual. Gender was inferred from the location of the ad webpage on the source website encoded in the ad urls. Contact information and names were discovered in the ad text itself. Images were analyzed as a way to further connect advertisers identified by contact information. The extracted data was stored in a MariaDB database for later analysis [27].

**Contacts to advertisers.** The main strategy for estimating population was to group ads together based on contact information. These groups of ads were considered the output of an *advertiser*, an entity which could represent one or more workers. For the purpose of identifying advertisers two main types of primary contacts were extracted: phone numbers and email addresses. Fig 2 illustrates the decision making process for accepting or rejecting contacts in ads.

Phone numbers were often in a numeric form, *416-555-1234* or similar. However, some phone numbers were obscured using combinations of numerals and words similar to *sevenseveneight 5five5 5421*. To identify advertisers, all phone numbers were converted into a common numeric form: *778 555 5421*. All extracted phone numbers were required to have valid North American area codes and were checked against the original ad to see if the phone could be matched. Any phone that could not be matched because it had been obscured was checked visually before being included as a contact for that advertiser.

Similarly, extracted emails could be misspelled or be extracted in error if the advertiser used the @ symbol in a way that might mimic an email address. Emails with poorly formed domains were visually inspected. Additionally, the Levenshtein distance [28] was calculated for every pair of emails. Email pairs with a Levenshtein distance of 2 or less were flagged for visual inspection. Groups of emails which appeared to be simple misspellings of each other were given a single canonical email address.

Most of the time advertisers could be identified by a single contact. However, in some cases ads could contain multiple contacts. To avoid overcounting, groups of co-occurring contacts were given a virtual contact identifier called a *cluster*. These clusters were identified using the DBSCAN algorithm ["pop/clusters.py" in 26,29]. Fig 3 illustrates a virtual contact *Cluster1* that contains the contacts *Phone1*, *Phone2*, and *Email1* in a group of ads. Contacts were considered related when they appeared together in at least one ad. Clusters are only meaningful in the context of a specific time period and are created as needed when population estimates are calculated. Any contact could either be stand alone or in exactly one cluster for any time period studied.

**Advertisers to people.** In the context of this study, advertisers could be thought of as a hidden variable represented by contacts. However, ultimately advertisers represent people. In

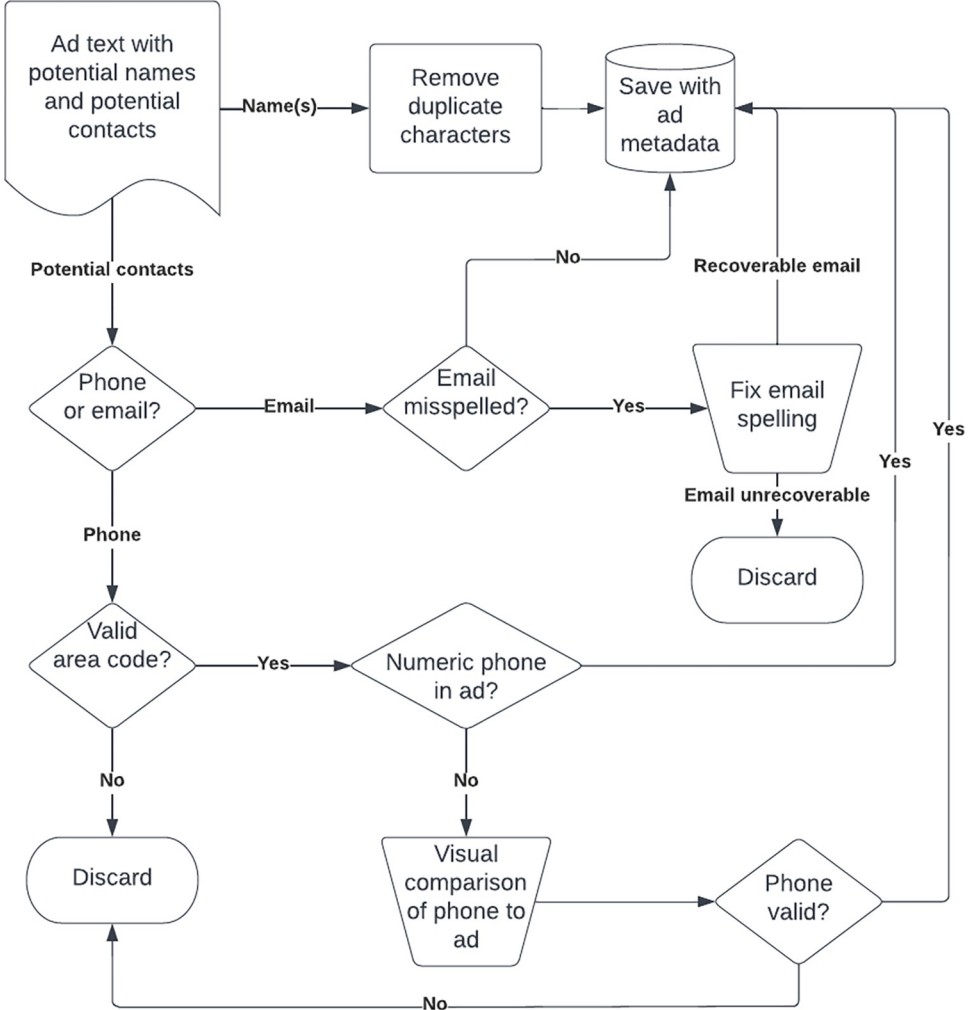

**Fig 2. Contact and name processing flow diagram.**

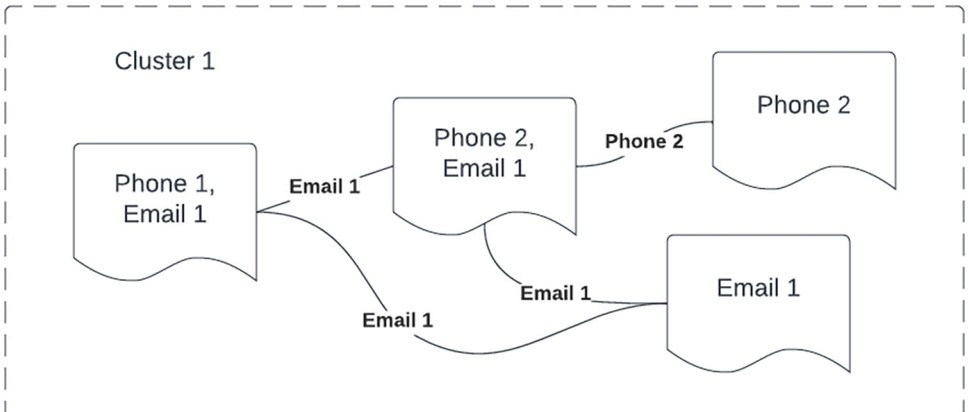

**Fig 3. Example cluster created by DBSCAN.** Ads are linked by common contacts. The virtual contact *Cluster1* is associated with the ads containing contacts *Phone1*,*Phone2* and *Email1*. The primary contacts *Phone1*, *Phone2* and *Email1* are considered "neighbors" because each contact appeared in at least one ad with another contact in the cluster.

order to determine approximately how many individuals an advertiser represented, first names were detected in ads. Most workers only used first names to identify themselves.

Detecting names in ads was a multi-step process of language model generation, refinement and finally application where a list of 1000 popular female and 1000 popular male first names [30] were compared against trigrams extracted from ads containing those name strings. Candidate name strings that appeared to be mostly used as names were retained. These seed names were used to identify words that typically preceded names (for example "name is . . ." or "je m'appelle . . ."). Trigrams containing these context words were then used to identify a larger set of names actually used in ads.

To count names in ads, name strings were identified in ad text where characters were converted to lowercase and all punctuation was removed except single quotes ["namelist" in 26,31]. As shown in Fig 2, when a potential name was found, it was saved in a canonical form with repeated letters removed. For example, a name "Angellaaaaaa" in the original text would be converted to the canonical form of "angela" before being stored. Names found in individual ads were collected and attached to the associated advertiser. Each canonical name used by an advertiser was treated as a unique worker for the purposes of population estimates.

An advertiser was considered a *collective* representing multiple individual workers if the advertiser either had more than one name associated with them or used keywords that indicated the advertiser represented a group of workers. The following keywords were used to detect collective advertisers missing name data: "models", "girls", "we", "our", "us", "spa", "agency", "club", "nous", "filles", "agence", "four hands", "duo", "trio", "roommate" and "couple".

**Image analysis.** Images were hashed with the perceptual hashing algorithm [32,33] to identify similar images. Common images could be used to detect when advertisers had changed contacts. Faces were detected using the Python *mtcnn* module to identify images with people [34] (see also S1 Appendix). Facial analysis was used as an aid in identifying workers when generating error parameters.

## Error detection and mitigation

**Error estimates.** The metadata extraction process could produce errors. Erroneous contacts were removed and co-occurring contacts were combined as described above. To further correct for overcounting, the probability that an advertiser represented contact sex workers, *P (a relevant)*, was estimated by visually inspecting ads from a random sample of advertisers using the criteria outlined in supplemental materials SI File. Secondly, *P(n valid)*, the probability that a name had been correctly extracted for a given advertiser, was estimated by visually inspecting a random sample of advertiser-name pairs.

**Calculating rates of contact change.** Population estimates could be inflated when an advertiser changed name or contact information. To measure these changes, sequences of ads were examined and frequencies of changed names and contacts were tallied ["pop/name-changed.pl" and "pop/idchanged.pl" in 26]. Face images in both cases were used as a proxy to identify an individual advertiser independently of name or contact. Ads with common face images but new contacts or names were considered changed.

Ad sequences used for detecting name change contained ads with at least one face image and only one contact per ad. For contact change, ad sequences from Site 3 were used. These advertisers could be identified independently with an internal chat id. The Site 3 ads all contained only one name and at least one face image. As Site 3 advertisers could change chat ids, the rate of chat id change was also measured by counting the number of chat ids associated with Site 3 contacts.

**Scaling calculation for advertiser and worker counts.** To mitigate the effects of errors and advertisers changing name and contact information, python modules were developed to adjust or *scale* population and advertiser counts for any time period ["pop" in 26]. A base module *pop.py* collected advertisers and names for a specific time period, generated clusters, added median name counts where names were missing and scaled back the counts based on the formulae described below. This base module was run repeatedly for different time periods by *multipop.py* to generate descriptive statistics.

Advertisers, who could represent one or more workers based on groups of ads, were estimated for a given period using Eq 1:

$$\hat{N}_{advertisers} = P(a\,relevant)\sum_{a\in Advertisers} P(a\,unique) \tag{1}$$

Where $\hat{N}_{advertisers}$ is the adjusted number of advertisers for a given period. *P(a relevant)* is the measured probability that any given advertiser is relevant. *Advertisers* is the original set of advertisers active during the period. *P(a unique)* is the probability that an advertiser had not changed contacts during this time.

*P(a unique)* was estimated using Eq 2:

$$P(a\,unique) = \frac{1}{1 + NewContacts} = \frac{1}{1 + \left(\frac{Period}{Days(a)} - 1\right) \cdot R_{idchange}} \tag{2}$$

Where *Period* is the total number of days in the period, *Days(a)* is the number of days that advertiser *a* was online during the period and *1 <= Days(a) <= Period*. The constant $R_{idchange}$ is the measured rate per day that an advertiser adds new contacts. If we assume that advertisers tend to advertise for the same length of time each time they advertise, the number of days where an advertiser may create a new contact would be approximately the ratio of the *Period* and *Days(a)* minus the one known period in the interval.

Sex worker population for a given period based on name counts was estimated using Eq 3:

$$\hat{N}_{workers} = P(n\,valid)P(n\,unique)P(a\,relevant)\sum_{a\in Advertisers} Names(a)P(a\,unique) \tag{3}$$

Where $\hat{N}_{workers}$ is the estimated sex worker population for a given period. *Names(a)* is the estimated name count found for advertiser *a* in the period or, if no names were found, the median number of names based on social context (individual or collective). *P(n valid)* is the measured probability that any detected name is valid for a given advertiser and *P(n unique)* is the measured probability that a name was not changed.

$R_{idchange}$ is defined by Eq 4:

$$R_{idchange} = R_{idchanges/day} + R_{chatidchange} \tag{4}$$

Where $R_{id\,changes/day}$ is the measured rate of new contacts per day and $R_{chatid\,change}$ is the measured rate of new chat ids per day for Site 3 advertisers.

The 95 percent confidence intervals for Formulas 1 and 2 were calculated by first finding the confidence intervals (*CI*) for the *P* and *R* parameters. The scaling calculations were then rerun using the lower and upper *CI* values for the input parameters. For the probabilities the *CI* was calculated using Eq 5 [35]:

$$CI = P \pm z\sqrt{P(1-P)/N} \tag{5}$$

Where *z* is the z function value for the 95 percent confidence interval, *P* is the parameter value and *N* is the sample size used to determine *P*. The root sum of squares, shown in Eq 6, was used

to calculate the confidence interval *CI* for the $R_{idchange}$ parameter that combined two rates:

$$CI = R_{idchange} \pm z\sqrt{R_{\text{id changes/day}}^2/N_{\text{id changes/day}} + R_{\text{chatid change}}^2/N_{\text{chatid change}}} \tag{6}$$

**Estimating advertisers using image data.** An alternate way to estimate advertisers uses image sharing. If we know how many images advertisers use on average and we assume that advertisers changing contacts use their own images and tend to use the same images in ads, advertiser counts can be estimated with Eq 7:

$$\hat{N}_{advertisers} = \frac{Uniqueimages}{Avimagesperadvertiser * R_{imagereused}} P(arelevant) \tag{7}$$

Where $\hat{N}_{advertisers}$ is the estimated number of advertisers for the duration of the study, *Unique Images* is the count of unique image hashes found associated with advertisers, *Av images per advertiser* is the average number of images used by any advertiser and $R_{image\ reused}$ represents the average number of times images were reused by advertisers and *P(a relevant)* is the measured proportion of relevant advertisers. A limitation of this technique is that the image parameters must be measured uniquely for each time period.

## Measures and statistical analyses

Advertiser population estimates were generated for the whole two year period as well as monthly and weekly. Advertiser population estimates consisted of two variables: the raw advertiser count and the scaled estimate from Eq 1. Biannual scaled advertiser estimates were also calculated using Eq 7. Descriptive statistics were generated for monthly and weekly estimates. Days online were estimated for all advertisers, calculated as the number of days between the first date and last date ads from that advertiser were seen.

Similarly, worker population estimates were generated weekly, monthly and for the two year period stratified by gender and social context (individual vs collective). These estimates consisted of three variables: the raw name count for the time period, the name count where advertisers missing names are assigned median name counts and a scaled estimate that corrects for advertisers changing contacts and names. Descriptive statistics were generated for monthly and weekly population counts. As a point of comparison with existing research, scaled population estimates for cis female workers were compared with the 2016 Canadian 20–49 year old female population from Statistics Canada [36].

Monthly trends were calculated for downloaded ads, advertisers and workers. These trends were considered for all workers as well as stratified by gender. The R *lm* function [37] was used to calculate univariate linear models between the number of months from the start of data collection (independent variable, range 0–25) and the dependent variables of downloaded ad count, adjusted advertiser count and adjusted worker count for each month.

## Verifying population estimates

The advertiser and worker population estimates were compared to four other data sources as well as an earlier analysis (see S2 Appendix). Firstly, the per-capita average weekly estimate of active Canadian workers of all genders was compared with a 2006 spot estimate for indoor workers in New Zealand [12]. Secondly, advertisers from non-classified sites during this period are compared with the advertisers identified from classified ads. Thirdly, the demographic breakdown of a large sample from Argento et al. [15] is compared with the composition of the advertisers identified in this study. Lastly, advertiser estimates for two periods:

November 1, 2014 to August 1, 2015 and November 1, 2015 to August 1, 2016 are compared to chat id counts from Site 3 collected between November 1, 2021 and August 1, 2022.

### Ethics statement

All source data used in this study consisted of publicly available data at the time it was collected and was collected in accordance with the policies of the sites in effect at the time. The methods used are conformant with the ethical standards of the Canadian Sociology Association (section 4.10 II) and the American Sociology Association (section 10.5 c) [38,39]. As the replicability of the main results of this paper is important, a data set is provided as part of the supporting information along with the code used to process it. However, in order to protect the safety and privacy of advertisers and third parties, all identifying information has been removed including the names of the source websites.

## Results

### Downloaded ads

A total of 3641544 web pages were collected from websites hosting Canadian contact sex work advertising between November 1, 2014 and December 31, 2016. The majority of the pages, 3545247 (97.36%), were collected from six classified ad sites designated here as Sites 1 to 6. The classified ad pages, where the advertiser and publication date was unambiguous, were used in the time based analysis. As a comparison, 96297 pages were downloaded from Canadian non-classified adult advertising sites over the same time period. Table 2 provides a breakdown of what was downloaded.

Advertisers who used classified advertising tended to only advertise on one classified site. Of the 6 websites, advertisers used on average 1.08 sites to advertise (standard deviation 0.3, median 1.0).

Contact information was not found in 399318 (11.0%) of the classified ads used in the 2014–2016 population estimates. A sample of 6955 ads representing 3975 unique cleaned ad texts were evaluated using the criteria in supplemental materials S1 file. The ads were judged to be relevant contact sex work ads 76.4% of the time (95% CI 75.4%-77.4%). Advertisers on average produced 17.8 ads (SD = 199.8). Assuming this rate for ads without contact information, 17139 +/- 0.5 advertisers (unscaled) may have been missed or an additional 9.3%.

### Error estimates

The probability that a name was valid for a given advertiser, *P(n valid)*, was 0.9612 +/- 0.0322 based on a random sample of 3415 unique advertiser-name pairs. The probability that a given

**Table 2. Web pages collected per source in 2014–2016.**

| Source | Pages collected | Percent |
|---|---|---|
| Site 1 | 851206 | 23.37% |
| Site 2 | 2057728 | 56.51% |
| Site 3 | 220071 | 6.04% |
| Site 4 | 409381 | 11.24% |
| Site 5 | 5832 | 0.16% |
| Site 6 | 1029 | 0.03% |
| Non classified sites | 96297 | 2.64% |
| **Total classified ads** | **3545247** | **97.36%** |
| **Total ads** | **3641544** | **100.00%** |

advertiser represented relevant contact sex workers, *P(a relevant)*, was 0.9500 +/- 0.0294 based on a random sample of 3999 advertisers.

Contacts were found to change much more frequently than names. Advertisers were estimated to change contacts at a rate of 0.0223 +/- 0.0017 contacts per day from a sample of 8454 ads representing 928 Site 3 advertisers, all of which had ads with images containing faces and only one chat id, name and contact represented in the ad. Ads with the same chat id and face images were considered to belong to the same advertiser.

As it was free to register on Site 3 it was relatively easy for advertisers to have more than one chat id. Counting the number of chat ids associated with 4761 single contacts on Site 3 showed that advertisers created 0.0024 +/- 0.0001 new chat ids per day. The rate of chat id change was combined with the rate of contact change to arrive at the estimated true rate of contact change in the calculation that scaled back the advertiser and name counts described in the scaling calculation section above.

The probability that a name had not been changed by an advertiser *P(n unique)* was 0.9953 +/- 0.0940. The probability was estimated based on a sample of 221865 ads representing 431 advertiser/name pairs. This sample was not restricted by site. All ads had only one contact and name and had images containing faces. A name was considered changed when the contact and face images had been reused with a different name.

## Population estimates

Fig 4 illustrates month to month trends during the 2014–2016 study period and Table 3 summarizes linear regression results between month and various population measures. Throughout this period there was a significant positive relationship between almost all measured statistics and month. Only new ads did not increase significantly. A dip in ad volume in November 2015 was the result of the downloaders being shut down for a two week period early in the month.

The estimated number of Canadian advertisers for the 2014–2016 study period was 75600 (95% CI 74087–77219) based on the scaling formula described in Eq 1. Canadian worker estimates were generated using Eq 3: an average of 16846 (SD 5858) workers of all genders were estimated to be active weekly, monthly this average increased to 26326 (SD 5481) and over the two year study period 169473 sex workers (95% CI 166870–172226) were estimated to be active at least once. Population counts for the 2014–2016 study period are summarized in Table 4 stratified by self-identified gender comparing the original raw counts, counts with missing name data interpolated and the scaled estimates that correct for overcounting. Advertisers were estimated to be active for a mean of 73.3 days (SD 151.8, median 14, IQR 1–58).

The majority group were female identified sex workers who were estimated to represent 83.6% of workers overall (N = 141669 95% CI 139469–143996). Scaled counts in any given week in 2014–2016 suggest that an average of 13575 (SD 4994) cis female sex workers were active in Canada. Monthly, the average was 21344 (SD 5028). The Canadian adult female population between 20 and 49 years of age in 2016 was 7205721 [40]. Thus the weekly average would represent 0.2% (one in 531), the monthly average 0.3% (one in 338) and the biannual estimate 2.0% (one in 51). Most workers self-identified as white (53%) based on Site 3 data (see S3 Appendix).

**Advertisers estimated from image reuse.** The estimated number of advertisers for 2014–2016 based on image reuse was calculated to be 69562 (95% CI 69085–70047) from Eq 7 where the number of *Unique images* was 1640209 the *Av images per advertiser* was 16 (SD 87), $R_{imagereuse}$ was 1.4 (SD 1.3) and *P(a relevant)* was measured to be 0.95. This is 92% (90%-94%) of the estimated 75600 (95% CI 74087–77219) advertisers for this period from Eq 1.

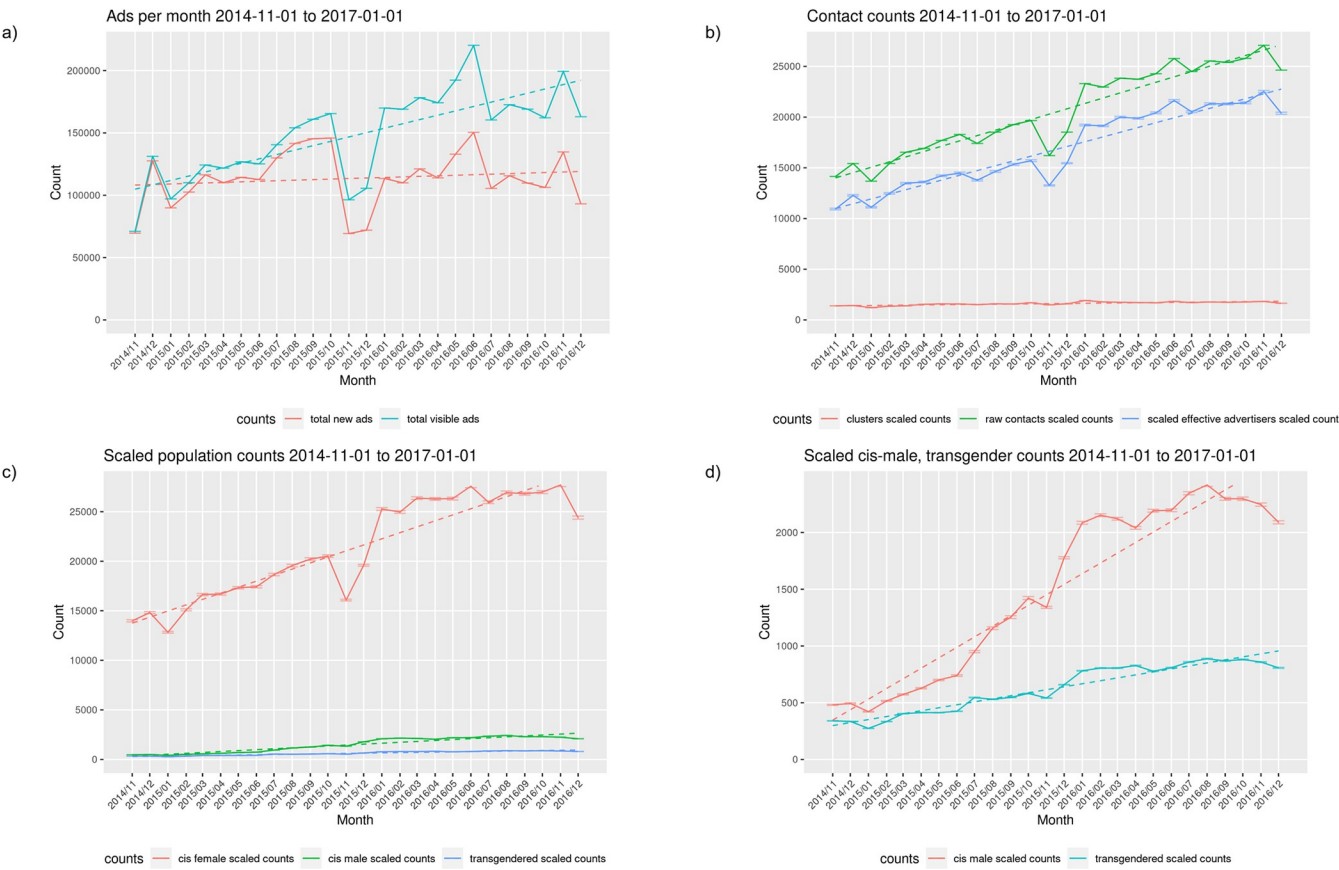

**Fig 4. Canadian monthly counts for the period 2014-11-01 to 2016-12-31.** a) ads downloaded; b) contacts, advertisers and clusters; c) estimated population by gender; d) male and transgendered population detail.

**Cluster analysis.** Ads with multiple contacts were treated as single advertisers using clustering as described above. Clusters could represent individuals or groups of workers. Weekly there was an average of 1013 clusters (SD 242), monthly this increased to an average of 1614 (SD 172). Over the 2014–2016 data collection period 12886 clusters were identified. Where clusters occurred they were typically not large. For the entire 2014–2016 period the median number of contacts represented by a cluster was 2 (IQR 2–3, mean 3.0, SD 7.5).

**Table 3. Changes by month from start of data collection for 2014–2016.**

| Variable | Beta parameter | Standard error | p value |
|---|---|---|---|
| Ad volume | 3501.5 | 620.9 | p<0.001 |
| New ads* | 439.4 | 588.9 | 0.463 |
| Scaled advertisers | 471.5 | 31.3 | p<0.001 |
| Clusters | 18.2 | 2.7 | p<0.001 |
| Scaled workers | 662.2 | 55.9 | p<0.001 |
| Scaled cis-female | 608.2 | 51.0 | p<0.001 |
| Scaled cis-male | 92.2 | 6.1 | p<0.001 |
| Scaled trans-female | 26.3 | 1.6 | p<0.001 |

*result not significant.

**Table 4. Name and advertiser counts for the period between 2014-11-01 and 2016-12-31.** Gender is based on ad category. Percentages are relative to total names. Corrected estimates are generated using Eq 3 for names and Eq 1 for advertisers.

| Category | Unscaled counts | Missing names added | Scaled (corrected) counts |
|---|---|---|---|
| Cis female names | 252133 (84.7%) | 258795 (83.5%) | 141669 (95% CI 139469–143996, 83.6%) |
| Cis male names | 15333 (5.1%) | 18141 (5.9%) | 8013 (95% CI 7843–8194, 4.7%) |
| Transgender names | 4931 (1.7%) | 5050 (1.6%) | 2943 (95% CI 2903–2985, 1.7%) |
| Other* names | 25407 (8.5%) | 27736 (8.9%) | 16849 (95% CI 16656–17051, 9.9%) |
| Total names | 297805 | 309924 | 169473 (95% CI 166870–172226) |
| Advertisers | 172767 | n/a | 75600 (95% CI 74087–77219) |

*refers to ads where no gender was indicated.

**Individual versus collective advertisers.** In 2014–2016, the unscaled number of collective advertisers was significantly smaller than the number of individual advertisers: 80040 versus 87499 respectively (collective proportion 47.8%, CI 47.5–48.0%, p<0.001). However, the unscaled name counts suggest that the number of workers who work collectively are significantly larger: 217847 collective versus 87499 individual (collective proportion 71.3%, CI 71.2–71.5%, p<0.001). The R *prop.test* [37] function was used to compare proportions.

Overall there was a median of 1 name per advertiser (IQR 1–2, average 1.8, SD 4.4). Collective advertisers had a median of 2 names per advertiser (IQR 1–3, average 2.4, SD 6.3). Most advertisers were associated with two names or less (unscaled N = 147434, 88%).

## Comparing the results with other data sources

**Comparison with a New Zealand population estimate.** Proportional to population, the estimated number of workers in this study is similar to estimates described in the New Zealand PRLC report from 2008 [8]. The New Zealand study did not distinguish workers based on gender. The male and female 20 to 49 year old population from the 2006 New Zealand census was 1777770 [41]. Of the 2396 workers counted in the report in February to March 2006, 2143 were off-street workers, representing 121 workers per 100000. The researchers took great care to ensure that workers represented in the study were in fact active at the time they were counted.

The weekly Canadian average reported here of 16846 (SD 5858) represents 122 (SD 43) workers per 100000 for the 20 to 49 year old population of 13761540 in 2016 [36]. A proportions z-test indicated no significant difference (p = 0.5) between the Canadian and New Zealand population estimates. While these proportions are tantalizingly close, we should be cautious in their interpretation. The time distance between the New Zealand and Canadian estimates may have an effect on population size. Both studies likely represent the majority of sex workers at the time they were conducted, however, neither this study nor the New Zealand study should be interpreted as an exhaustive census.

**Comparison with non-classified advertising.** Advertisers who did not use classified advertising but nevertheless had an online presence were rare. Advertisers who used non-classified websites numbered 1550 or 1.0% of all advertisers (unscaled count). Of these, 1028 or 0.7% of all advertisers (unscaled), were found to exclusively use non-classified advertising sites.

**Comparison with Argento et al..** Argento et al. [15] provides a detailed demographic breakdown of a large sample of participants (N = 852). Study participation was limited to cis and trans women from the Vancouver, BC area. Sexual minority participants were much more prevalent in Argento et al. at 36.3% (N = 309) compared to the 1.7% trans women represented

in the scaled population counts found here. Argento et al. did not distinguish between different types of gender nonconformity. Indigenous participants were also much more prevalent at 38.8% (N = 331) compared to the 1.3% represented in the Site 3 data (see S3 Appendix). Primarily street-involved participants represented 50.7% (N = 432) of the sample. Of the 420 off street workers, the R *prop.test* function [37] showed that, similar to the advertising data, significantly more worked in a collective context: 253 or 60.2% (CI 55.4–64.9%, p<0.001).

**Comparison with Site 3 data from 2021–2022.** After 2016, Site 3 appears to have become the dominant advertising site for sex workers in Canada. Data collection was resumed using the same methods described above in October 2021 and is ongoing. The raw count of Site 3 advertisers, directly measured using chat id metadata from November 1, 2021 to July 31, 2022 was 48832. The advertisers used an average of 1.1 chat ids (SD 0.6) based on a sample of 2605 advertisers who used single phone numbers. The proportion of relevant advertisers was 0.68 (2707 from a sample of 4000 advertisers) based on criteria outlined in supplemental materials S1 File. Thus, the actual number of advertisers for this period is estimated to be 30042.8 (95% CI 30042.5–30043.0).

Between November 1, 2014 and August 1, 2015 there were an estimated 33145 advertisers (95% CI 32670–33654). Between November 1, 2015 and August 1, 2016 the number of advertisers grew to 47361 (95% CI 46766–47990). The 2021–2022 estimate is 90.6% and 63.4% of the 2014–2016 estimates respectively. Most likely the COVID-19 pandemic will have affected the number of active contact sex workers in 2021–2022.

## Discussion

The purpose of this study was to gain insight into how sex worker populations change over time. Ad web pages from sites commonly used by sex workers were downloaded from November 1, 2014 to December 31, 2016. Analysis of primary contacts in the ads identified advertisers and analysis of first names in ads identified individual workers associated with these advertisers. Advertisers were active for a mean 73.3 days (SD 151.3, median 14, IQR 1–58) and 88% represented two workers or less. Population estimates generated weekly, monthly and over two years showed that the number of advertisers and workers increased significantly as the length of the analysis period increased, providing evidence that workers frequently enter and exit the industry. Comparisons with other data sources suggest that the metadata extraction and scaling techniques used are plausible on both short and long time scales. However, the demographic stratification represented in the advertising data does not appear to match that found in a recent qualitative study [15] suggesting that non-random sampling strategies used in qualitative research may not accurately reflect the greater Canadian sex worker population.

### The effect of time

The element of time turned out to be crucial for interpreting population estimates. Even after controlling for changing contact information and co-occurring contacts, recently active workers were part of a much larger cohort of workers only intermittently active. The average estimated number of workers active week to week represented one tenth the estimated population for the whole two year period. This order of magnitude difference may seem surprising. However, in the context of the Canadian economy it is plausible.

If financial stress is a motivation for entry into sex work there are a very large number of women living in poverty in Canada, in 2015 this was estimated to be 14.7% [42]. Sex work, where it intersects with poverty, may represent one of many informal survival strategies [12,43–45]. The brevity of involvement for most advertisers indicates sex work was likely not permanent employment for most workers. However, it would be a mistake to assume that all

sex workers are economically disadvantaged. Prior research shows that there is wide variation in what sex workers earn [12,43,44] (see also S4 Appendix). Other types of informal workers frequently do not require their informal income to survive [46,47] and this may be true for many sex workers. The demographic patterns described in this study lend support for the view, also extensively reported in the literature, that Canadian sex workers exercise a substantial degree of agency in how they engage with the industry even in the face of structural obstacles [16,18,20,48–51].

Research sponsored by the New Zealand Prostitution Law Reform Committee (PLRC) [7,12], some of which is described above, further illustrates how population counts can be misinterpreted when the dimension of time is not taken into account. Four estimates were generated as part of this research. Two were generated before the enactment of the Prostitution Reform Act of 2003 (PRA) based on NGO and police statistics: 8000 workers from the New Zealand Prostitutes Collective (NZPC) and 5932 workers based on police records. Two were generated after based on a direct enumeration of workers: 2396 in February-March 2006 and 2332 in June-October 2007 [8,12]. The pre-2003 counts, based on cumulative statistics, overestimated the number of active workers because many workers were included who had already left the industry.

## Implications for research and policy

How representative are the samples used in the large body of Canadian qualitative research? Benoit and Shaver [52] note that this is difficult to determine when the characteristics of the greater population are unknown. Sample sizes can be small: 10 samples used in 13 studies [14,15,19,21–24,43,44,53–55] ranged from 21 to 852 (median 206.5, IQR 65.25–461.75, mean 288, SD 291.3). Based on the scaled population estimates, a minimum random sample of 206 cis female workers would be required to achieve a 5% confidence interval at a 95% confidence level [56]. Only three studies [14,15,23] had more than the minimum cis female participants. However, none of these studies used random sampling. Adding to the problem, Canadian research often shares participants between studies. For example, *An Evaluation of Sex Workers Health Access* (AESHA), a growing cohort of cis and trans female workers from Vancouver, BC, is shared by three studies [14,15,23]. Another sample of 218 participants from six Canadian municipalities is shared by five studies [17,19,21,53,54] including a working paper from 2014 that is the source of the often quoted statistic that Canadian sex workers are in the industry for an average of 10 years [54].

Non-random samples can be misleading even when large. For example, Argento et al. [15], which had the largest sample overall (N = 852), describe their participants as "highlighting the overrepresentation of gender and sexual minorities and Indigenous women among sex workers in Vancouver". This statement is not consistent with the demographic makeup of the much larger group of online advertisers. For example, the proportion of Indigenous women to the Canadian female population is 4.8% based on the 2016 census [57] however the proportion of advertisers who self-identified as Indigenous in 2014–2016 was 1.3% (see S3 Appendix) indicating that Indigenous people are likely underrepresented in the industry.

In contrast, trans people are likely to be overrepresented but not to the extent indicated by Canadian research. A census test conducted in 2019 by Statistics Canada [58] found the proportion of trans people at that time to be 0.35%: far less than the 1.7% estimated here. In contrast, the AESHA cohorts had very high proportions of LGBTQ2S participants (25.3% to 36.3%) but made no distinction between trans and other gender nonconformity making direct comparison difficult. Studies that explicitly identified trans participants [17,43,44,55], while proportionally fewer than the AESHA cohorts, also had participation much greater than 1.7%

(mean 7.8%, SD 5.15%). Proportions of men in Canadian research (mean 17.2%, SD 0.6%) were also higher than the 4.7% of workers estimated here [17,43,44].

Canadian studies often track workers over multiple years [14,15,23,43] potentially giving the impression that the majority will require help in exiting the industry. However, prior research shows that many workers find such offers of assistance intrusive [8,59]. This study supports this perspective. Advertising for more than one year was very uncommon and, as described in S4 Appendix, long-term individual advertisers routinely take breaks from advertising.

If the majority of sex workers only have sporadic involvement in the industry, what are they likely to need from policy? Because these workers are not well represented in research, this question is difficult to answer. More research specifically targeted at these workers is needed to determine what may be appropriate.

### Directions for future research

The clustering algorithm used to disambiguate co-occurring contacts revealed that, over time, workers can participate in large scale social networks. The largest cluster found covered almost the entire country and was estimated to represent over 800 people; how pervasive is this type of ad hoc collective activity? Secondly, an analysis of advertiser restrictions showed that 16.89% of advertisers restrict clients based on skin color; what is motivating this choice?

Online classified ad data can be used as the basis for integrated qualitative research as metadata, once identified, can be used to stratify samples for further investigation. Parallelizing metadata extraction with data collection is important as advertisers are typically transient and timely contact is essential. Even a cursory review of existing research shows that integration of these research streams is needed to ensure that qualitative samples are representative and conversely to ensure that archival metadata is accurate.

### Limitations

The population estimates presented in this study should not be interpreted as an exhaustive census of sex workers in Canada during 2014–2016. Instead the scaled estimates, extracted from the very large selection of classified ads collected, most likely represent a lower bound on the actual population.

It is possible that the collected ads may be an incomplete set. The list of sites used as starting points may not have been complete and metadata used to quantify advertisers and workers was not always available. We should also remember that, while other venues became less prominent during the study period [60], not all workers use online advertising. Furthermore, not all online advertising was usable for population counts. Issues with other online sources included not knowing if co-occurring contacts were from related advertisers and whether advertisers were active at the time the data was collected.

Estimates of contact change based on images have limitations: advertisers changing both contacts and images at the same time could be missed and advertisers that reuse images from other unrelated advertisers could be erroneously included. It is not possible to mitigate these sources of error from archival data alone. Other hard to detect sources of error were: advertisers using multiple identities simultaneously and workers changing work contexts without leaving the industry. Relevant qualitative research is needed to help resolve these questions.

### Conclusions

This study is believed to be the first to consider sex worker population in the context of long term advertising behavior in an industrialized democracy using online archival sources. The

data presented suggests that most workers are likely only active for brief periods of time. While there may be more than one reason for this pattern we must consider that the majority of workers in fact exercise agency and have autonomy in how they practice sex work. The fact that most workers were advertising in a collective context does not diminish this possibility as in general these appear to be small ad hoc collectives similar to the Small Owner Operated Brothel (SOOB) model described in the New Zealand research [12].

Intermittently active sex workers considered over long periods of time represent a much larger population than what one would expect from worker estimates from shorter time periods. Demographic studies must take into account the duration of the data collection period and how long any individual worker was active in that period. Similarly, to avoid misleading results, the demographic composition of non-random samples should be situated in the spectrum of workers actually active for the time period being considered.

The online advertising space provides opportunities to engage with sex workers in ways that may not have been possible before. The challenge for researchers, policy makers and advocacy groups will be to ensure that underrepresented groups are included in any discussion on sex workers' future conditions of work.

## Supporting information

**S1 Appendix. Image validation.**
(DOCX)

**S2 Appendix. Comparison with the SPACES study.**
(DOCX)

**S3 Appendix. Multiple populations.**
(DOCX)

**S4 Appendix. Factors affecting advertiser longevity.**
(DOCX)

**S1 File. Criteria for deciding relevance.**
(XLSX)

**S2 File. Detailed spreadsheet with breakout of days online by demographic category.**
(XLSX)

**S3 File. Detailed spreadsheet with breakout of number of ads by demographic category.**
(XLSX)

## Author Contributions

**Conceptualization:** Lynn Kennedy.

**Data curation:** Lynn Kennedy.

**Formal analysis:** Lynn Kennedy.

**Funding acquisition:** Lynn Kennedy.

**Investigation:** Lynn Kennedy.

**Methodology:** Lynn Kennedy.

**Project administration:** Lynn Kennedy.

**Resources:** Lynn Kennedy.

**Software:** Lynn Kennedy.

**Supervision:** Lynn Kennedy.

**Validation:** Lynn Kennedy.

**Visualization:** Lynn Kennedy.

**Writing – original draft:** Lynn Kennedy.

**Writing – review & editing:** Lynn Kennedy.

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
