## [Decision Letter · Decision Letter 0]

18 May 2022

PONE-D-22-00964The Silent Majority: Evidence for Part Time Sex Work in CanadaPLOS ONE

Dear Dr. Kennedy,

Thank you for submitting your manuscript to PLOS ONE. After careful consideration, we feel that it has merit but does not fully meet PLOS ONE’s publication criteria as it currently stands. Therefore, we invite you to submit a revised version of the manuscript that addresses the points raised during the review process.

Thanks so much for submitting your work to PLOS ONE.

The manuscript was evaluated by two rviewers and you can find the comments below.

Please provide these items in the revised version.

- Please provide the aim of the study cleary in the abstract and full text. The purpose should be written at the end of the introduction of the abstract and full text.

- The conclusion should be in the context of the results.

- Introduction is too long and could be shorten.

- Please provide the ethical consideration and ethics code if available.

-The number of Tables and Figures are too high. If possible, please merge some tables and also you can move some of the tables or Figures to the appendix.

- Please provide the limitations of the study to the discussion.

We look forward to receiving your revised manuscript.

Kind regards,

Hamid Sharifi

Academic Editor

PLOS ONE

Journal Requirements:

2. In your Methods section, please include additional information about your dataset and ensure that you have included a statement specifying whether the collection and analysis method complied with the terms and conditions for the source of the data.

Reviewers' comments:

Reviewer's Responses to Questions

**Comments to the Author**

1. Is the manuscript technically sound, and do the data support the conclusions?

Reviewer #1: Partly

Reviewer #2: Yes

2. Has the statistical analysis been performed appropriately and rigorously? 

Reviewer #1: I Don't Know

Reviewer #2: I Don't Know

3. Have the authors made all data underlying the findings in their manuscript fully available?

Reviewer #1: No

Reviewer #2: Yes

4. Is the manuscript presented in an intelligible fashion and written in standard English?

Reviewer #1: Yes

Reviewer #2: Yes

5. Review Comments to the Author

Reviewer #1: 1. Summary of the research

The authors conducted a study on the dynamics of contact sex work in Canada, how the population changed over time and representation of sex workers in policy debate and research used for policy. They found the population of sex workers to be dynamic, with most sex workers having only brief involvement, while more of them are inactive than those active. The authors concluded that as policy in Canada is based on qualitative research using small, self-selecting samples, the perspectives of these short-term workers, who are the silent majority, may not be adequately represented.

The authors show that though most previous studies were large scale using multiple techniques, they generated population estimates at a single point in time, focusing on female sex workers as subjects. Few studies tried to directly estimate sex work population demographics from publicly available internet advertising data. Because of the methodologies used by those studies, the estimates are difficult to confirm and tend to overestimate the population of sex workers. Using online data not only includes populations enumerated using the previous methodologies but also encompasses the more transient sex workers usually left out, who advertise online. This study therefore adds value to the current body of knowledge as it improves accuracy and completeness of other methodologies to shed more light on the part time and transient sex workers in Canada who are actually the majority.

Though there are issues with the study methodology that need attention, the authors did a good job of describing their methodology and how the data was analyzed, including parsing metadata, contacts and images, and analyzing other variables to describe sex worker advertising behavior. The methods section however needs to be rewritten to ensure it flows better and addresses flaws highlighted below.

The results are also presented in detail though some tables and description of the results could be improved. The authors were able to bring out the element of time which turned out to be key to interpreting population estimates. Limitations were well described with follow up actions.

My overall recommendation is that the study needs major revisions, especially the methods section, with some reanalysis once some of the methodological issues area addressed.

2. Examples and evidence

2.1. Major issues

2.1.1. The abstract introduction does not fully to summarize the context. From the third sentence (line 8 to line 11), the source of the information is unclear. If it is from literature, the reference should be provided. In the event it is from the current study, these would be results which should be better moved to the results section.

2.1.2. The Methods section needs a lot of work as it does not flow well, does not describe the methodology well and includes results (e.g. lines 112 – 114). I think that it needs some major reorganization and rewrite. I suggest that lines 112-114 be moved to the results section. The methods could then start with a description of the identification of the web sites to be used as the source of data, then explaining why the 6 sites were selected (part of this is available in lines 116 – 120).

2.1.3. In paragraph 2, starting on line 121, the authors write that the downloaded ads should be considered as a sample. There is need to explore and explain to what extent the downloaded ads are representative of all ads on the site. Without this being addressed, it is not clear how the authors can be sure that the findings of the study can be generalizable.

2.1.4. In the description of the contents of table 2 (end of line 127 to start of line 129), the reasons for excluding some sections in sites 2 and 3, other than to simplify analysis and reduce irrelevant posts (though it’s not clear to what extent) are not clear to me. I believe this may actually have introduced sampling bias. I would therefore suggest that the excluded section be reintroduced, and analysis done again.

2.1.5. The authors explain in line 135-136 that because the study uses publicly available data, there is no need for ethical approval. In the next sentence starting in line 136, they however report that there is personal identifying information in the data which they had to remove. I agree that it is not necessary to get consent from the participants, which would otherwise be very difficult if not impossible. However, I believe that because of the presence of personal identifying information in the data, the authors should seek ethical approval for the study.

2.1.6. The first section under results, starting from the sentence in line 227 up to line 237, seems to be describing methods. I would suggest that this information be moved to the methods section.

2.1.7. With regards to the tables presented in the manuscript, in general, I believe that they are too many and there are a number of issues that need address. I will give examples of issues with some of the tables and actions that can be considered. Table 1 does not contain adequate information to justify its inclusion, while other tables, like table 8, are too complex and will need to either be simplified, be converted to a graphic presentation or be removed altogether and the data described in the text. Some tables are too busy and difficult to read (e.g. table 7 and especially Table 10 such that it is difficult to fully understand the message they are meant to convey. The categories may need to be regrouped and/or reorganized. In Table 10, including disparate groups like individual, male, French, escort, etc, in the same column may be comparing dissimilar variables. On the other hand, some tables (e.g. Table 1) have very scanty information to the extent that it is not clear what they are meant to communicate. Some, like table 15, have excessively long titles with too much information which could preferably by moved to the text, while others like Table 1 are too brief and need to be made more descriptive. Table 12 seems to have a wrong titles while others have headers that need review and revision, e.g. table 13, in which the last column header is called “names”, but the information contained in the column is actually numbers. For table 16, the reference in the text (line 348) seems to be referring to another table and needs correction. Finally, some tables are broadly referred to in the text, but their contents are not summarized in the text, for example tables 1, 3, 5, 7 and 8.

2.1.8. The evidence on how the industry regulates itself through economics of supply and demand (presented in line 425-478) from table 16 may be more appropriately presented in the results section. The interpretation of the evidence could however still be presented in the discussion section.

2.1.9. The change mentioned in the sentence in lines 249-250 that reads “A change on Site 1 which concealed contact information resulted in few contacts being extracted after June 2015” may affect the findings of the change in sex worker population dynamics over time. The effect of this change may need to be taken into account in the analysis and interpretation of relevant results.

2.1.10. In the sub-section on the effect of time (line 427-434), many of the figures presented were not described in the results section. They therefore seem misplaced and should be moved to the results. Their interpretation is what may be more relevant to be included in the discussion section.

2.1.11. In the discussion section, with regards to the implication of policy (ln 465 – 515), I feel that there is inadequate data presented. More analysis could be done from the data used to construct table 16 and more data could be presented in the results section to give a good background for discussion. With little data presented on the implications of policy, I feel that there is a bit of overreach with regards to the conclusions and it will help to present a bit more data to adequately support the conclusions.

2.2. Minor issues

2.2.1. I feel that the title does not adequately convey the key features of the article, though it does spark interest. I suggest that “part time” in the title be replaced with something about the transient nature of sex work in Canada, and other components such as online advertising, long term behavior of sex workers (since it is the first study to consider the long-term behavior of sex workers based on online advertising).

2.2.2. The sentence in line 114 starting with “Table 1 outlines...” is very vague and needs to highlight what was collected. The table itself may need to be moved to the results section.

2.2.3. The statement in line 133-134 that reads “Advertisers with no valid ads had all their ads checked for relevance” is not clear to me. My assumption is that if they had no valid ads, they should be excluded. It can be deleted.

2.2.4. Table 8 and Table 10 look very busy and I would suggest reducing the number of columns by putting the standard deviation in brackets in the same cell as the average days. The number of decimal points could also be harmonized and reduced possibly to 1 decimal point. The median IQR could also be added, and be put in the same cell in brackets after the relevant median day values.

2.2.5. Fig 1 is mentioned in lines 245-247, but the summary of the findings is not included in the text, and the text needs to be updated accordingly.

2.2.6. Under limitations (line 411), the relevance of the statement “At least four additional studies could be written based on this archival data” is not clear to me. I would suggest that it be deleted.

2.2.7. In the conclusion, with regards to the sentence starting in line 517, it is not clear what techniques are being referred to. It may be better for the authors to expound on what they are referring to.

Reviewer #2: It is a very important study, but we need to be sure that the people included are sex workers, so I think that the selection process should have a peer review to be sure that the same results are reached.

6. PLOS authors have the option to publish the peer review history of their article (what does this mean?). If published, this will include your full peer review and any attached files.

Reviewer #1: **Yes: **Dr Brian Chirombo (MBChB, MPH)

Reviewer #2: **Yes: **Edgard J. Narvaez D.

---

## [Author Response · Author response to Decision Letter 0]

2 Sep 2022

PONE-D-22-00964

The Silent Majority: Evidence for Part Time Sex Work in Canada

PLOS ONE

Dear Dr. Kennedy,

Thank you for submitting your manuscript to PLOS ONE. After careful consideration, we feel that it has merit but does not fully meet PLOS ONE’s publication criteria as it currently stands. Therefore, we invite you to submit a revised version of the manuscript that addresses the points raised during the review process.

Thanks so much for submitting your work to PLOS ONE.

The manuscript was evaluated by two rviewers and you can find the comments below.

Please provide these items in the revised version.

- Please provide the aim of the study cleary in the abstract and full text. The purpose should be written at the end of the introduction of the abstract and full text.

Added the following to the abstract: “The purpose of this study is to consider how time affects the population dynamics of contact sex workers in Canada using publicly available internet advertising data collected over multiple years.”

- The conclusion should be in the context of the results.

moved some material from the conclusion to the discussion

- Introduction is too long and could be shorten.

shortened the introduction to less than 600 words and moved some material to the discussion.

- Please provide the ethical consideration and ethics code if available.

added the following ethics statement: “All source data used in this study consisted of publicly available data at the time it was collected and was collected in accordance with the policies of the sites in effect at the time. The methods used are conformant with the ethical standards of the Canadian Sociology Association (section 4.10 II) and the American Sociology Association (section 10.5 c) [34,35]. As the replicability of the main results of this paper is important, a data set is provided as part of the supporting information along with the code used to process it. However, in order to protect the safety and privacy of advertisers and third parties, all identifying information has been removed including the names of the source websites.”

-The number of Tables and Figures are too high. If possible, please merge some tables and also you can move some of the tables or Figures to the appendix.

Reduced the number of tables in the main paper to 4. 

Removed two figures but added 3 figures (for a total of 4) to the methods to better illustrate the data processing pipeline, how contacts were handled and how the clustering algorithm for contacts works.

- Please provide the limitations of the study to the discussion.

moved and expanded the limitations to the end of the discussion.

If applicable, we recommend that you deposit your laboratory protocols in protocols.io to enhance the reproducibility of your results. Protocols.io assigns your protocol its own identifier (DOI) so that it can be cited independently in the future. For instructions see: https://journals.plos.org/. Additionally, PLOS ONE offers an option for publishing peer-reviewed Lab Protocol articles, which describe protocols hosted on protocols.io. Read more information on sharing protocols at https://plos.org/protocols?.

We look forward to receiving your revised manuscript.

Kind regards,

Hamid Sharifi

Academic Editor

PLOS ONE

Journal Requirements:

https://journals.plos.org/ and

https://journals.plos.org/

2. In your Methods section, please include additional information about your dataset and ensure that you have included a statement specifying whether the collection and analysis method complied with the terms and conditions for the source of the data.

This has been done.

3. We note that you have indicated that data from this study are available upon request. PLOS only allows data to be available upon request if there are legal or ethical restrictions on sharing data publicly. For more information on unacceptable data access restrictions, please see http://journals.plos.org/.

This is no longer the case. linked an anonymized data set in the supplemental materials.

This is no longer an issue.

b) If there are no restrictions, please upload the minimal anonymized data set necessary to replicate your study findings as either Supporting Information files or to a stable, public repository and provide us with the relevant URLs, DOIs, or accession numbers. For a list of acceptable repositories, please see http://journals.plos.org/.

Data is available here: https://osf.io/mebvp/

Data was shared with the University of British Columbia as part of the SPACES project as described in the paper with no direct involvement in the project. If you could provide some clarification on how to handle this situation that would be much appreciated. 

Comments to the Author

1. Is the manuscript technically sound, and do the data support the conclusions?

Reviewer #1: Partly

Reviewer #2: Yes

2. Has the statistical analysis been performed appropriately and rigorously?

Reviewer #1: I Don't Know

Reviewer #2: I Don't Know

3. Have the authors made all data underlying the findings in their manuscript fully available?

Reviewer #1: No

Reviewer #2: Yes

4. Is the manuscript presented in an intelligible fashion and written in standard English?

Reviewer #1: Yes

Reviewer #2: Yes

5. Review Comments to the Author

Reviewer #1: 1. Summary of the research

The authors conducted a study on the dynamics of contact sex work in Canada, how the population changed over time and representation of sex workers in policy debate and research used for policy. They found the population of sex workers to be dynamic, with most sex workers having only brief involvement, while more of them are inactive than those active. The authors concluded that as policy in Canada is based on qualitative research using small, self-selecting samples, the perspectives of these short-term workers, who are the silent majority, may not be adequately represented.

The authors show that though most previous studies were large scale using multiple techniques, they generated population estimates at a single point in time, focusing on female sex workers as subjects. Few studies tried to directly estimate sex work population demographics from publicly available internet advertising data. Because of the methodologies used by those studies, the estimates are difficult to confirm and tend to overestimate the population of sex workers. Using online data not only includes populations enumerated using the previous methodologies but also encompasses the more transient sex workers usually left out, who advertise online. This study therefore adds value to the current body of knowledge as it improves accuracy and completeness of other methodologies to shed more light on the part time and transient sex workers in Canada who are actually the majority.

Though there are issues with the study methodology that need attention, the authors did a good job of describing their methodology and how the data was analyzed, including parsing metadata, contacts and images, and analyzing other variables to describe sex worker advertising behavior. The methods section however needs to be rewritten to ensure it flows better and addresses flaws highlighted below.

The results are also presented in detail though some tables and description of the results could be improved. The authors were able to bring out the element of time which turned out to be key to interpreting population estimates. Limitations were well described with follow up actions.

My overall recommendation is that the study needs major revisions, especially the methods section, with some reanalysis once some of the methodological issues area addressed.

This has been done. The data was reanalyzed as recommended. Comparisons were also completed to see how population estimates derived from advertising relate to other estimates reported in the literature. 

2. Examples and evidence

2.1. Major issues

2.1.1. The abstract introduction does not fully to summarize the context. From the third sentence (line 8 to line 11), the source of the information is unclear. If it is from literature, the reference should be provided. In the event it is from the current study, these would be results which should be better moved to the results section.

 The lines 8 - 11 have been removed and the purpose of the study was added instead

2.1.2. The Methods section needs a lot of work as it does not flow well, does not describe the methodology well and includes results (e.g. lines 112 – 114). I think that it needs some major reorganization and rewrite. I suggest that lines 112-114 be moved to the results section. The methods could then start with a description of the identification of the web sites to be used as the source of data, then explaining why the 6 sites were selected (part of this is available in lines 116 – 120).

This section has been extensively revised. Some sections have been moved to results.

The sites should be considered participants in the study along with the advertisers and the individual people they represent. The sites have been de-identified because of the legal situation in Canada vis-a-vis advertising for sex work at the moment (the Protection of Communities and Exploited Persons Act). 

Added flow diagrams to show the data gathering and extraction process which is intended to make the processing steps clearer.

Added a section on sources of error and how they were mitigated.

Added a section on techniques to validate the collected data by comparing it with other research results as well as techniques to check internal validity.

2.1.3. In paragraph 2, starting on line 121, the authors write that the downloaded ads should be considered as a sample. There is need to explore and explain to what extent the downloaded ads are representative of all ads on the site. Without this being addressed, it is not clear how the authors can be sure that the findings of the study can be generalizable.

Explained in more detail how often ads were searched and indicated that all new ads on the sites were downloaded at least every 15 minutes. This download frequency was in fact necessary to keep up with the large volume of ads published.

Also included an analysis of data that was not included in the analysis as a form of comparison. 

Note that even if the set of ads is not complete we are not in danger of overcounting the potential number of workers - which turns out to be quite large in any case.

2.1.4. In the description of the contents of table 2 (end of line 127 to start of line 129), the reasons for excluding some sections in sites 2 and 3, other than to simplify analysis and reduce irrelevant posts (though it’s not clear to what extent) are not clear to me. I believe this may actually have introduced sampling bias. I would therefore suggest that the excluded section be reintroduced, and analysis done again.

This has been removed and a reanalysis of the complete data set was done instead. The results are basically unchanged.

2.1.5. The authors explain in line 135-136 that because the study uses publicly available data, there is no need for ethical approval. In the next sentence starting in line 136, they however report that there is personal identifying information in the data which they had to remove. I agree that it is not necessary to get consent from the participants, which would otherwise be very difficult if not impossible. However, I believe that because of the presence of personal identifying information in the data, the authors should seek ethical approval for the study.

External ethics approval is not possible at the moment as the study is not being conducted in an institutional context. Great care has been taken to ensure that participants’ privacy is respected and that ethics guidelines have been followed.

The SPACES study received ethics approval from the University of British Columbia to use the data collected. The SPACES investigators did not do any data collection but instead used a subset of the data described in this study (see Appendix D). All data was from publicly available sources at the time of collection, and although a reasonable expectation of privacy may not apply in this case as persons observed were advertising to the public, layers of anonymization were enforced in the publicly released data to discourage data linkage.

Updated the ethics statement (see above). Data was collected in a way that is conformant to the ethics guidelines of the American Sociology Association and the Canadian Sociology Association.

2.1.6. The first section under results, starting from the sentence in line 227 up to line 237, seems to be describing methods. I would suggest that this information be moved to the methods section.

This has been removed as it exists already in other parts of the paper.

2.1.7. With regards to the tables presented in the manuscript, in general, I believe that they are too many and there are a number of issues that need address. I will give examples of issues with some of the tables and actions that can be considered. Table 1 does not contain adequate information to justify its inclusion, while other tables, like table 8, are too complex and will need to either be simplified, be converted to a graphic presentation or be removed altogether and the data described in the text. Some tables are too busy and difficult to read (e.g. table 7 and especially Table 10 such that it is difficult to fully understand the message they are meant to convey. The categories may need to be regrouped and/or reorganized. In Table 10, including disparate groups like individual, male, French, escort, etc, in the same column may be comparing dissimilar variables. On the other hand, some tables (e.g. Table 1) have very scanty information to the extent that it is not clear what they are meant to communicate. Some, like table 15, have excessively long titles with too much information which could preferably by moved to the text, while others like Table 1 are too brief and need to be made more descriptive. Table 12 seems to have a wrong titles while others have headers that need review and revision, e.g. table 13, in which the last column header is called “names”, but the information contained in the column is actually numbers. For table 16, the reference in the text (line 348) seems to be referring to another table and needs correction. Finally, some tables are broadly referred to in the text, but their contents are not summarized in the text, for example tables 1, 3, 5, 7 and 8.

Removed most of the tables from the main text and updated the text to describe the tables that remain. 

Created appendices for the more detailed information on how demographic variables affect advertising behavior. included the original excel spreadsheets for the two tables summarizing this behavior as supplemental materials. added IQR to these spreadsheets.

2.1.8. The evidence on how the industry regulates itself through economics of supply and demand (presented in line 425-478) from table 16 may be more appropriately presented in the results section. The interpretation of the evidence could however still be presented in the discussion section.

Revised these paragraphs and, after reviewing the correlations between the actual number of charges vs the estimated number of advertisers (in contrast to the figures provided which use per capita measurements) have removed the province to province comparison. Generally the number of charges is correlated with the number of advertisers (pearson correlation 0.84, p < 0.001). Why this is the case is a topic for future research as under the PCEPA workers cannot be charged with an offense.

2.1.9. The change mentioned in the sentence in lines 249-250 that reads “A change on Site 1 which concealed contact information resulted in few contacts being extracted after June 2015” may affect the findings of the change in sex worker population dynamics over time. The effect of this change may need to be taken into account in the analysis and interpretation of relevant results.

Despite missing some data, we still see an increasing population of workers as time scale increases while weekly spot estimates remain in the range of 15-20k workers. The fact that we know some data is missing will not change this effect as the majority of the data was intact. 

2.1.10. In the sub-section on the effect of time (line 427-434), many of the figures presented were not described in the results section. They therefore seem misplaced and should be moved to the results. Their interpretation is what may be more relevant to be included in the discussion section.

These have been moved to the results section as requested.

2.1.11. In the discussion section, with regards to the implication of policy (ln 465 – 515), I feel that there is inadequate data presented. More analysis could be done from the data used to construct table 16 and more data could be presented in the results section to give a good background for discussion. With little data presented on the implications of policy, I feel that there is a bit of overreach with regards to the conclusions and it will help to present a bit more data to adequately support the conclusions.

See the response to 2.1.8.

2.2. Minor issues

2.2.1. I feel that the title does not adequately convey the key features of the article, though it does spark interest. I suggest that “part time” in the title be replaced with something about the transient nature of sex work in Canada, and other components such as online advertising, long term behavior of sex workers (since it is the first study to consider the long-term behavior of sex workers based on online advertising).

Changed the title. Generally in Canada the view espoused in the press, derived from qualitative research that almost universally uses non-random samples, is that most sex workers work for 10 years in the industry but this study shows that that this view is too simplistic, thus the title reflects this: our beliefs are likely wrong. What is really needed are studies that use better sampling techniques.

2.2.2. The sentence in line 114 starting with “Table 1 outlines...” is very vague and needs to highlight what was collected. The table itself may need to be moved to the results section.

This has been updated.

2.2.3. The statement in line 133-134 that reads “Advertisers with no valid ads had all their ads checked for relevance” is not clear to me. My assumption is that if they had no valid ads, they should be excluded. It can be deleted.

This has been removed for clarity. The approach being taken now is to start with all advertisers then reduce the number based on a measured number that were shown to not be advertising contact sex work. The corrected population estimates reflect this. The actual number of excluded advertisers works out to be 5%. 

2.2.4. Table 8 and Table 10 look very busy and I would suggest reducing the number of columns by putting the standard deviation in brackets in the same cell as the average days. The number of decimal points could also be harmonized and reduced possibly to 1 decimal point. The median IQR could also be added, and be put in the same cell in brackets after the relevant median day values.

These have been replaced with excel spreadsheets as described above and have been removed from the main text to the appendices.

2.2.5. Fig 1 is mentioned in lines 245-247, but the summary of the findings is not included in the text, and the text needs to be updated accordingly.

Added a summary of the findings to the text.

2.2.6. Under limitations (line 411), the relevance of the statement “At least four additional studies could be written based on this archival data” is not clear to me. I would suggest that it be deleted.

 Removed this. 

2.2.7. In the conclusion, with regards to the sentence starting in line 517, it is not clear what techniques are being referred to. It may be better for the authors to expound on what they are referring to.

 This has been removed from the conclusions and expanded and added to the discussion. The techniques being referred to are the data collection and metadata extraction techniques described in the paper. The hope is that the expanded content will make this clearer.

The issue, as described above, with much of the qualitative research in Canada is that it depends on non-random samples. This issue can be partly mitigated by using advertisers as a basis for random samples. 

Reviewer #2: It is a very important study, but we need to be sure that the people included are sex workers, so I think that the selection process should have a peer review to be sure that the same results are reached.

Peer review of the source sites was done as part of the SPACES project. All of the sites selected were recommended by industry insiders who used them. An expanded section is added to the methods to emphasize this. Both Site 1 and Site 2 have been the subject of other research (Boekner et al. for example).

Because of privacy issues regarding potentially identifying workers, the peer review has been limited to the original members of the SPACES team. However, Appendix D is added which compares the data presented here with the SPACES investigators’ preliminary results. The results in this study are very conservative in comparison as the current project undertakes additional error control measures to minimize overcounting.

Included in the supplemental materials S1 File are the criteria used to decide if ads were related to contact sex work.

The analysis of image data provides another way to validate the advertisers. The 1752880 unique images found for an estimated 70-75000 advertisers suggests these are indeed real people. See Appendix A for an analysis of the images.

To improve external validity the population estimates are compared with other studies and a more recent data set from 2021-2022. This more recent data set is available in anonymized form on the supplemental materials site.

Ultimately, the hope is that other researchers will attempt to replicate the results. That would be a very positive outcome.

6. PLOS authors have the option to publish the peer review history of their article (what does this mean?). If published, this will include your full peer review and any attached files.

Do you want your identity to be public for this peer review? For information about this choice, including consent withdrawal, please see our Privacy Policy.

Reviewer #1: Yes: Dr Brian Chirombo (MBChB, MPH)

Reviewer #2: Yes: Edgard J. Narvaez D.

---

## [Decision Letter · Decision Letter 1]

17 Oct 2022

PONE-D-22-00964R1The silent majority: the typical Canadian sex worker may not be who we thinkPLOS ONE

Dear Dr. Kennedy,

Thank you for submitting your manuscript to PLOS ONE. After careful consideration, we feel that it has merit but does not fully meet PLOS ONE’s publication criteria as it currently stands. Therefore, we invite you to submit a revised version of the manuscript that addresses the points raised during the review process.

We look forward to receiving your revised manuscript.

Kind regards,

Hamid Sharifi

Academic Editor

PLOS ONE

Journal Requirements:

Reviewers' comments:

Reviewer's Responses to Questions

**Comments to the Author**

1. If the authors have adequately addressed your comments raised in a previous round of review and you feel that this manuscript is now acceptable for publication, you may indicate that here to bypass the “Comments to the Author” section, enter your conflict of interest statement in the “Confidential to Editor” section, and submit your "Accept" recommendation.

Reviewer #1: All comments have been addressed

Reviewer #2: All comments have been addressed

2. Is the manuscript technically sound, and do the data support the conclusions?

Reviewer #1: Yes

Reviewer #2: Yes

3. Has the statistical analysis been performed appropriately and rigorously? 

Reviewer #1: I Don't Know

Reviewer #2: Yes

4. Have the authors made all data underlying the findings in their manuscript fully available?

Reviewer #1: Yes

Reviewer #2: Yes

5. Is the manuscript presented in an intelligible fashion and written in standard English?

Reviewer #1: Yes

Reviewer #2: Yes

6. Review Comments to the Author

Reviewer #1: The authors have adequately addressed my comments raised in the previous round of review.

However, the revised manuscript has some new minor issues that I believe the authors need to address. These issues being minor however only entail minor revision of the manuscript.

1. The sentence starting in line 109 that begins with "Issues with these other source ....." could be better moved to the discussion section.

2. In Table 3 (from line 349), the authors present p values less than 0.001 as decimal numerals. However, the standard convention is for such p values to be expressed as p<0.001 regardless of the actual decimal numeral. I would therefore suggest that the authors consider following the standard convention and present all the p values that are less than 0.001 as p<0.001.

3. In lines 388, 390 and 427, the authors present Odds Ratios (ORs) without the corresponding confidence intervals (CIs). I would suggest that they consider presenting all ORs with their corresponding CIs as per standard convention.

Reviewer #2: Comments and questions raised in the previous revision have been incorporated or clarified.

The methodology is clear although a bit extensive.

7. PLOS authors have the option to publish the peer review history of their article (what does this mean?). If published, this will include your full peer review and any attached files.

Reviewer #1: **Yes: **Brian C Chirombo, MBChB, MPH

Reviewer #2: No

---

## [Author Response · Author response to Decision Letter 1]

26 Oct 2022

PONE-D-22-00964R1

The silent majority: the typical Canadian sex worker may not be who we think

PLOS ONE

Dear Dr. Kennedy,

Thank you for submitting your manuscript to PLOS ONE. After careful consideration, we feel that it has merit but does not fully meet PLOS ONE’s publication criteria as it currently stands. Therefore, we invite you to submit a revised version of the manuscript that addresses the points raised during the review process.

We look forward to receiving your revised manuscript.

Kind regards,

Hamid Sharifi

Academic Editor

PLOS ONE

Journal Requirements:

 • All references were checked for retractions. No retractions were found as of 2022-10-17.

Reviewers' comments:

Reviewer's Responses to Questions

Comments to the Author

1. If the authors have adequately addressed your comments raised in a previous round of review and you feel that this manuscript is now acceptable for publication, you may indicate that here to bypass the “Comments to the Author” section, enter your conflict of interest statement in the “Confidential to Editor” section, and submit your "Accept" recommendation.

Reviewer #1: All comments have been addressed

Reviewer #2: All comments have been addressed

2. Is the manuscript technically sound, and do the data support the conclusions?

Reviewer #1: Yes

Reviewer #2: Yes

3. Has the statistical analysis been performed appropriately and rigorously?

Reviewer #1: I Don't Know

Reviewer #2: Yes

4. Have the authors made all data underlying the findings in their manuscript fully available?

Reviewer #1: Yes

Reviewer #2: Yes

5. Is the manuscript presented in an intelligible fashion and written in standard English?

Reviewer #1: Yes

Reviewer #2: Yes

6. Review Comments to the Author

Reviewer #1: The authors have adequately addressed my comments raised in the previous round of review.

However, the revised manuscript has some new minor issues that I believe the authors need to address. These issues being minor however only entail minor revision of the manuscript.

1. The sentence starting in line 109 that begins with "Issues with these other source ....." could be better moved to the discussion section.

 • This has been moved to the Discussion > Limitations section paragraph 2.

2. In Table 3 (from line 349), the authors present p values less than 0.001 as decimal numerals. However, the standard convention is for such p values to be expressed as p<0.001 regardless of the actual decimal numeral. I would therefore suggest that the authors consider following the standard convention and present all the p values that are less than 0.001 as p<0.001.

 • These have been changed to use the convention.

3. In lines 388, 390 and 427, the authors present Odds Ratios (ORs) without the corresponding confidence intervals (CIs). I would suggest that they consider presenting all ORs with their corresponding CIs as per standard convention.

 • The main reason for the calculation was two-fold:

 a. Show that just using advertisers as a measure of collective vs individual can be misleading.

 b. Show that, while the Argento et al. sample was different on some measures, it was comparable to the advertising data on the collective vs individual dimension when considering indoor workers.

 • The comparisons were redone using the R prop.test function. The text has been updated to reflect the results. 

 • Previously, the odds ratio was calculated by dividing the probability of an advertiser or name being found in a collective context with that of the same being found in an individual context. For example, for advertisers this would be:

 Odds Ratio = p(adv is collective)/p(adv is individual) = (Collective Adv/All Adv)/(Individual Adv/All Adv) = (Collective Adv)/(Individual Adv)

As these are simple frequency counts, there is no confidence interval. In retrospect, prop.test seems to be more informative.

Reviewer #2: Comments and questions raised in the previous revision have been incorporated or clarified.

The methodology is clear although a bit extensive.

 • Because this is the first time this has been attempted, the methods are necessarily more detailed. In the future, if related work is being described, it is hoped that the Methods in this paper can be used as a reference.

---

## [Editor Report · Decision Letter 2]

31 Oct 2022

The silent majority: the typical Canadian sex worker may not be who we think

PONE-D-22-00964R2

Dear Dr. Kennedy,

We’re pleased to inform you that your manuscript has been judged scientifically suitable for publication and will be formally accepted for publication once it meets all outstanding technical requirements.

Kind regards,

Hamid Sharifi

Academic Editor

PLOS ONE
---

## [Editor Report · Acceptance letter]

5 Nov 2022

PONE-D-22-00964R2 

The silent majority: the typical Canadian sex worker may not be who we think 

Dear Dr. Kennedy:

I'm pleased to inform you that your manuscript has been deemed suitable for publication in PLOS ONE. Congratulations! Your manuscript is now with our production department. 

Kind regards, 

on behalf of

Dr. Hamid Sharifi 

Academic Editor

PLOS ONE